# N-Rich, Polyphenolic Porous Organic Polymer and Its In Vitro Anticancer Activity on Colorectal Cancer

**DOI:** 10.3390/molecules27217326

**Published:** 2022-10-28

**Authors:** Sabuj Kanti Das, Snehasis Mishra, Krishna Das Saha, Debraj Chandra, Michikazu Hara, Amany A. Mostafa, Asim Bhaumik

**Affiliations:** 1School of Materials Sciences, Indian Association for the Cultivation of Science, 2A & 2B Raja S. C. Mullick Road, Jadavpur, Kolkata 700032, India; 2Cancer Biology and Inflammatory Disorder Division, CSIR-Indian Institute of Chemical Biology, Jadavpur, Kolkata 700032, India; 3World Research Hub Initiative, Institute of Innovative Research, Tokyo Institute of Technology, Nagatsuta-cho 4259, Midori-ku, Yokohama 226-8503, Japan; 4Laboratory for Materials and Structures, Institute of Innovative Research, Tokyo Institute of Technology, Nagatsuta-cho 4259, Midori-ku, Yokohama 226-8503, Japan; 5Nanomedicine & Tissue Engineering Laboratory, Department of Ceramic, National Research Centre, El Bohouth St., Dokki, Cairo 12622, Egypt

**Keywords:** porous polymer, polyphenolic amine linked, high surface area, DNA damage, anticancer activity

## Abstract

N-rich organic materials bearing polyphenolic moieties in their building networks and nanoscale porosities are very demanding in the context of designing efficient biomaterials or drug carriers for the cancer treatment. Here, we report the synthesis of a new triazine-based secondary-amine- and imine-linked polyphenolic porous organic polymer material TrzTFPPOP and explored its potential for in vitro anticancer activity on the human colorectal carcinoma (HCT 116) cell line. This functionalized (-OH, -NH-, -C=N-) organic material displayed an exceptionally high BET surface area of 2140 m^2^ g^−1^ along with hierarchical porosity (micropores and mesopores), and it induced apoptotic changes leading to high efficiency in colon cancer cell destruction via p53-regulated DNA damage pathway. The IC30, IC50, and IC70 values obtained from the MTT assay are 1.24, 3.25, and 5.25 μg/mL, respectively.

## 1. Introduction

The word ‘cancer’ is ominous and saving lives from this deadly disease is a big challenge. The scientific community has paid huge attention over the years to solve this critical issue [1,2]. However, the goal is far away and continuous research efforts are underway for the detection of cancer at its earlier stages, as well as midway through its treatment [1,2]. Cancer is caused by uncontrollable cell growth in different body parts. Throughout the world, one in six deaths is caused by cancer [3]. In 2020, the World Health Organization (WHO) reported that even under the COVID-19 pandemic situation all over the world, 10 million people died from cancer, and there were 10.1 million new cases of common cancers [3,4]. Among the different types of cancer, such as blood, breast, colon, oral, skin, prostate, ovarian, bone marrow cancers, etc., colon cancer is very common for both men and women. A total of 935 thousand patients died from colon and rectum cancer in 2020 [3,4]. Some common symptoms give an indication of colon cancer: unexplained bleeding from the colon, changes in bowel habits, unusual body-weight loss, colon polyps, etc. [5,6,7]. Medicinal; surgical; chemo-, biological, and radiation therapy; cold atmospheric plasma (CAP) treatment, etc., are well-known treatments and are very often employed to treat colorectal carcinoma [8]. However, most of the time, cancer treatment deals with the destruction of affected cells through chemo- and radiotherapy, resulting in a serious probability of multiorgan dysfunctionality, nervous system damage, and the interruption of cellular equilibrium [9,10].

In the field of cancer research, different types of natural products or their derivatives, such as organic, inorganic, and organic–inorganic hybrid, nanocomposite materials, etc., are used as drug delivery vehicles, as well as potential drugs [11,12,13]. Porous nanomaterials possess high specific surface areas and tunable porosities; thus, they have huge scope for surface functionalization through immobilization and post-synthetic modifications [14,15,16]. Due to the presence of different reactive functional groups in the skeleton, the metal–organic frameworks (MOFs), covalent organic frameworks (COFs), porous organic polymers (POPs), and related organic–inorganic hybrid porous nanomaterials often displayed good anticancer activity [17,18,19,20,21]. Problems associated with large-scale synthesis, solvent stability, and metal leaching are the serious drawbacks [22] of MOF materials in the field of biomedical applications. Porous organic polymers are formed via covalent bond formation between the monomeric building units, which give these materials high chemical stability in different organic solvents. Furthermore, chemical robustness of the networks is needed for various potential applications [23,24,25]. Covalently bonded networks of POPs are utilized in various biomedical applications because of their cell penetration compatibility, cell viability, and large-scale synthesis facilities [26,27,28]. Anticancer activity studies have been carried out on a colorectal carcinoma cell line (HCT 116), where these porous nanomaterials act as cell destroyer or drug delivery systems [29,30,31]. Functionalized POP materials having small particle sizes mean they are effective for target-specific cell death. There are several naturally occurring polyphenolic flavonoids, such as quercetin, bromelain, curcumin, epigallocatechin gallate (EGCG), etc., and they have therapeutic effects/potential against different types of cancer, such as colorectal carcinoma. Qi et al. reported an alternating consumption of quercetin and β-glucon that considerably reduced the mortality rates of mice with colon cancer [32]. Polyphenolic moiety present in quercetin is responsible for its good anticancer activity, and this has motivated us to explore the POP-bearing polyphenol and N-rich functionalities [33] built inside the structure. Apart from these functional groups, the particle sizes of these POPs are suitable for cell membrane permeability; therefore, they are considered as very promising candidates for a chemotherapeutic agent.

Herein, we demonstrated the successful synthesis of a new N-rich polyphenolic porous triazine-based organic polymer (TrzTFPPOP, Figure 1) through a simple Schiff base polycondensation reaction between 1,4-bis(4,6-diamino-s-triazin-2-yl)-benzene (SL-1) [34] and triformylphluoroglucinol [35]. Its anticancer activity on human colorectal carcinoma (HCT116) cell line has been explored.

## 2. Results and Discussion

Phenolic-OH and N-rich TrzTFPPOP has been synthesized through the Schiff base polycondensation reaction between tetra-amine and trialdehyde. The material TrzTFPPOP was thoroughly characterized by using different instrumental techniques. The presence of different types of bonding connectivities in the material was initially investigated from the FTIR spectrum, taking the transmittance from 4000 to 400 cm^−1^ as the synthesized monomers as well as that of the POP materials (Figure 1a). An aldehydic >C=O stretching band at 1666 cm^−1^ along with an -NH_2_ deformation peak between 1650 and 1657 cm^−1^ are attenuated, which confirms that the condensation reaction takes place between the aldehyde- and amine-containing monomer units. The characteristic peak at 1543 cm^−1^ is observed in both SL-1 ligand as well as TrzTFPPOP, indicating the presence of a triazine unit in them. An IR stretching band at 1623 cm^−1^ along with two broad peaks at 3226 and 3346 cm^−1^ are present, which suggested the presence of imine (-C=N) linkage, -OH, and aminal (-CH-NH-) groups, respectively [34]. The FTIR spectra of TrzTFPPOP samples after introducing deionized (DI) water, acid 2(M), and base 2(M) treatment (7 days) remained unchanged, which suggested structural the integrity of the polymer network (See Appendix A).

For the further confirmation of the polymerization process, we have performed the solid-state ^13^C CP MAS NMR analysis of the TrzTFPPOP material, and this is shown in Figure 1b. The downfield NMR signals appeared at 171,168, and 165 ppm, and they could be responsible for the presence of carbon atoms in the triazine ring as well as the carbon of the imine bond adjacent to the phenyl ring. On the other hand, the NMR peaks at 148, 138, and 110 ppm indicate the presence of different aromatic carbon atoms. NMR peaks at 148, 138, and 110 ppm indicate the different aromatic carbon atoms adjacent to the phenolic-OH, triazine-N rings, and aliphatic -CH_2_ group, respectively. NMR signals appeared at 53 and 129 ppm due to the presence of the aliphatic carbon and carbon atoms of the phenyl ring. The above solid-state ^13^C-NMR spectrum analysis confirmed the formation of the porous organic polymer material TrzTFPPOP through the formation of both imine and secondary aminal bonding linkages.

The crystalline nature of the material was investigated by collecting the powder X-ray diffraction pattern of the TrzTFPPOP material from 4 to 40 degrees of 2θ, which is shown in Appendix A (See Appendix A). The amorphous nature of the material was confirmed by the observed broad peak between 2θ values of 20 to 30 degrees, which is a characteristic PXRD pattern of amorphous material. Porosity and surface area analysis provided the information regarding the architectural rigidity, specific surface area, and permanent porosity of the porous organic material TrzTFPPOP. According to IUPAC nomenclature, the N_2_ adsorption/desorption isotherm of TrzTFPPOP (Figure 1c) can be classified as a combination of two types of isotherm, where a type I isotherm is followed by a steady increase in N_2_ uptake in the entire P/P_0_ region [36,37]. At the low-pressure region, a steep increase in the N_2_ uptake for the isotherms is observed, which indicates the presence of microporosity in the polymer network.

The gradual increase in the N_2_ uptake suggested the presence of mesopores originating from the interparticle void spaces. The BET surface area of the TrzTFPPOP material was calculated from the N_2_ adsorption/desorption isotherm, which was 2140 m^2^ g^−1^ with a total pore volume of 1.91 cc g^−1^. The pore size distribution (PSD) of TrzTFPPOP was estimated by using non-local density functional theory (NLDFT) on the N_2_ adsorption–desorption isotherms, and this is shown in Figure 1d. The observed pore widths of the material were 1.16, 2.73, and 4.5 nm, which suggested the presence of hierarchical micro- and mesoporosity in the polymer network. This high BET surface area and wide-scale porosity along with organic functionality (phenolic -OH, secondary amine, imine) could make the TrzTFPPOP material a potential candidate to counterattack the affected colorectal cancer cells. HR-TEM images of the powder samples before and after cell treatment have been conducted for the morphological analysis of the material, and these are represented in Figure 2a–d. The HRTEM images suggested self-assembled nanowire-like particles with average diameters of 8–10 nm. This small POP particle size is suitable for cell membrane penetration [38]. After the cell treatment, TrzTFPPOP was recovered and HRTEM images were collected, which are very similar to the as-synthesized material, indicating the structural and morphological integrity of the material. The thermal stability of the porous polymer network has been analyzed by performing TG analysis at 10 °C/min ramp under air in the temperature range of 25–800 °C, which is shown in Figure 3a. The TGA profile diagram revealed that the organic polymer network of TrzTFPPOP is thermally stable up to 308 °C; furthermore, we observed temperature structural degradation beyond that. The first weight loss up to around 100 °C could be attributed to the evaporation of surface-absorbed moisture. The chiral cationic POP material synthesized by Zhang et al. [39] also exhibited similar thermal stability, which is essential for exploring their applications under harsh environments. The solid-state UV–Vis spectrum of the material confirmed its visible-light absorption capability (Figure 3b). Nanomaterials with anticancer activity are currently being intensively studied as nanoparticles with sizes less than 100 nm, which can easily penetrate through the cell and selectively target the cancer cells without affecting other parts of the body [40]. Currently, nano-conjugates and submicron particles are progressively used as drug carriers to cells. Their entry into the cell epitomizes the preliminary step in this process, being most of the nanoparticles taken up by endocytosis [40]. Accordingly, we have measured the cytotoxicity of TrzTFPPOP in the human colorectal carcinoma (HCT 116) cell line by MTT assay.

A wide range of concentrations (0–50 µg/mL) of TrzTFPPOP was initially chosen and then the concentration scale was limited to 0–10 μg/mL for MTT assay. The MTT assay data are represented in Appendix A. MTT assay data suggested the IC30, IC50, and IC70 as 1.24, 3.25, and 5.25 μg/mL, respectively, after 12 h of TrzTFPPOP treatment. Thus, the concentration of 3 μg/mL of TrzTFPPOP was selected for detailed biological experimentation as this concentration of TrzTFPPOP had a 50% inhibitory potential of cell viability. The most impactful drug for colorectal cancer is 5-Fluorouracil (5FU), and the IC50 of 5FU is ~10 μg/mL. In the case of our synthesized TrzTFPPOP, the IC50 is lower than 5FU. We have also checked the cytotoxicity of TrzTFPPOP in a normal human epithelial kidney cell line (HEK293), where the IC50 is 12–13 μg/mL. The MTT assay data are represented in Appendix A. Thus, we observed 3.25 μg/mL, which is not toxic for healthy cells. The experiment was performed in triplicate and the values are represented as the mean ± SEM. Therefore, we have established TrzTFPPOP as a potent anticancer porous organic polymer, which has better efficacy than 5FU against HCT 116. This result has been further confirmed from the quantification of apoptosis and necrosis by Annexin V-FITC and PI staining after time-dependent (12 h and 24 h) treatment with TrzTFPPOP. The dose of TrzTFPPOP was selected as 3 μg/mL, and the IC50 of TrzTFPPOP (Figure 4A) followed the previous literature [41].

The externalization of phosphatidylserine is the hallmark signature for the cell death of apoptosis [42]. Annexin V-FITC binds to phosphatidylserine when it is present on the outer leaflet of the plasma membrane. Flow cytometric analysis represented the live (Q3), early apoptotic (Q4), late apoptotic (Q2), and necrotic (Q1) cell population. The cell population of Q3, Q4, Q2, and Q1 is 81.1, 1.5, 1, and 16.4, respectively, whereas after 12 h and 24 h treatment of 3 μg/mL TrzTFPPOP, the cell population of Q3, Q4, Q2, and Q1 is changed to 39.1, 23.2, 17.4, and 20.2, and 15.7, 0.7, 25.6, and 58, respectively. It has been observed that the cell population of Q4, Q2, and Q1 has been increased after 12 h treatment of 3 μg/mL TrzTFPPOP, while the Q3 population has been decreased after 24 h, and the late apoptotic and necrosis populations have increased. This flow cytometric data confirmed that cell death is characterized by apoptosis. After confirmation of apoptosis, the percentage of DNA content in each phase of the cell cycle was quantified to understand the cell-cycle arrest. Time-dependent treatment of 3 μg/mL TrzTFPPOP was completed. Then, the percentage of DNA content was quantified, and it was found that at G2/M phase the DNA content increased with time compared with the control DNA. Therefore, this flow cytometric data assured us that G2/M arrest is the cause of apoptosis (Figure 4B).

DNA damage often results in cell-cycle arrest via p53/p21 pathways. The flow cytometric data were obtained from BD LSRFortessa™ equipped with FlowJo version 10 software. Histone H2AX phosphorylated on serine 139 (γ-H2AX) is the major hallmark of DNA damage and DNA double-strand breaks (DSB). Thus, to evaluate the role of TrzTFPPOP in the initiation of DNA damage, we performed an immunofluorescence study to identify the phosphorylation of γ-H2AX underlying the foci formation in the nucleus. A significant number of γ-H2AX foci were found when the cells were treated with 3 μg/mL TrzTFPPOP. The number of foci increased with time. This data confirmed that TrzTFPPOP could involve the apoptosis mechanism through G2/M arrest via the phosphorylation of γ-H2AX (Figure 5). The statistical mean fluorescence intensity was also plotted via Graphpad Prism Software.

A DNA double-strand break was induced by the phosphorylation of p53 at Serine15. Therefore, we have checked the expression of P-p53. We have also checked the expression level of p21. The confocal image showed that the FITC and PE expression increases in a time-dependent manner. The statistical mean fluorescence intensity was also plotted via Graphpad Prism Software. This result confirmed that the expression level of P-p53 and p21 increased after treatment with 3 μg/mL TrzTFPPOP (Figure 6). The apoptotic changes include blebbing, cell shrinkage, nuclear fragmentation, chromatin condensation, and chromosomal DNA fragmentation, which confirmed the TrzTFPPOP induced apoptosis. The presence of polyphenolic moieties in the chemical structure of the TrzTFPPOP porous organic polymer could be responsible for this programmed cell death.

## 3. Materials and Methods

### 3.1. Chemicals and Cell Lines

We obtained terephthalonitrile and dicyandiamide from Sigma-Aldrich, Balgalore, India. We procured phloroglucinol, hexamine(hexamethylenetetramine), and trifluoroacetic acid from Spectrochem, Mumbai, India. We purchased KOH from Merck, Bangalore, India. We used organic solvents, viz., dimethylsulphoxide (DMSO), 2-methoxyethanol, THF, MeOH and acetone, as received from Spectrochem, India. We obtained cell culture media components, viz., Dulbecco’s Modified Eagle Medium (DMEM), Penicillin–Streptomycin–Neomycin (PSN), antibiotic cocktail, fetal bovine serum (FBS), ethylenediaminetetraacetic acid (EDTA), and trypsin from Gibco, USA. We obtained other compulsory fine and raw chemicals from SRL, Mumbai, India, and Sigma-Aldrich, USA. We purchased human colorectal carcinoma (HCT 116) and human epithelial kidney (HEK293) cell line from National Centre for Cell Sciences (NCCS), India. We procured antibodies were procured from Cell Signaling Technology (CST), USA, and eBioscience, USA. We obtained other reagents from best existing commercial sources and performed all reactions without further purification.

### 3.2. Material Characterizations

We collected FTIR spectra of the as-synthesized, different-solvent-treated TrzTFPPOP and monomers by PerkinElmer Spectrum 100 spectrophotometer (Boston, MA., USA). We used a 500 MHz Bruker Advance III spectrometer (Karlsruhe, Germany) to record solid-state ^13^C CP MAS NMR spectrum of POP material at a MAS frequency of 10 kHz. We collected the wide-angle PXRD patterns of TrzTFPPOP from Bruker D-8 Advance SWAX diffractometer (Karlsruhe, Germany) using Cu Kα (λ = 0.15406 nm) radiation. After washing with plenty of DI water and different organic solsvent, we dried the powder material in a hot air oven at 160 °C for 12 h. Then, we activated the powder materials at 150 °C under high vacuum to obtain guest-free material for sorption analysis. We used Quantachrome Autosorb 1-C surface area analyzer (Boynton Beach, FL., USA) to obtain N_2_ adsorption/desorption isotherms at 77 K. We applied non-local density functional theory (NLDFT) on the N_2_ adsorption/desorption isotherms to calculate pore size distribution (PSD) of the activated material. We carried out morphological studies by JEOL JEM 2010 transmission electron microscope (Tokyo, Japan). We used TA-SDTQ-600 thermogravimetric analyzer (TGA) of TA Instruments, New Castle, DE, USA with a scanning rate of 10 °C/min in air to obtain the thermal analysis profile diagram. We carried out solid-state UV–Vis spectrum analysis by using Shimadzu UV-2401PC (Tokyo, Japan). We have collected ^1^H and ^13^C NMR spectra using 400 MHz Bruker AVANCE II spectrometer (Karlsruhe, Germany). We carried out absorbance studies of different solutions by ELISA reader (Model: Emax, Agilent-Thermo Fisher, Tokyo, Japan, at 595 nm). We calculated the proportions of live, apoptotic, and necrotic cells by flow cytometry (BD LSRFortessaTM, San Jose, CA, USA) [43]. We examined staining images of the double immunofluorescence using confocal laser scanning microscope (FV 10i, Olympus, Tokyo, Japan) [44].

### 3.3. Synthesis of 1,4-Bis(4,6-diamino-s-triazin-2-yl)-benzene (SL-1)

Following a reported procedure [45] with slight modification, we took terephthalonitrile (512.5 mg, 4.0 mmol), dicyandiamide (672.6 mg, 8.0 mmol), and KOH (78.5 mg, 1.4 mmol) along with 12 mL of 2-methoxyethanol solvent in a microwave (G-30) vial. We sealed and placed the reaction mixture into the MW reaction chamber at 195 °C under autogenerous pressure for 10 min (Figure 1). After cooling down the reaction mixture to 25 °C temperature, we poured the content into a beaker containing hot water. Then, we filtered off the white-colored precipitate that formed and washed with water followed by methanol and acetone to achieve pure compound SL-1 (75% yield). We characterized the pure product by collecting 1H NMR (400 MHz) and 13C NMR (100) data, in addition to 1H NMR in DMSO-d6: δ = 8.33 (s, 4H, Ar); 6.80 (br s, 8H, NH2) and 13C NMR in DMSO-d6: δ = 169.71, 167.40, 139.43, 127.39 ppm.

### 3.4. Synthesis of Triformylphloroglucinol (TFP)

We followed a procedure from the literature [46] with little modification, and put anhydrous phloroglucinol (6.014 g, 49 mmol), hexamethylenetetramine (15.098 g, 108 mmol), and 90 mL of trifluoroacetic acid into a pre-dried 250 mL two-neck RB with a magnetic stir bar, which maintained an inert atmosphere. We heated the reaction mixture for 3 h at 100 °C followed by the addition of 150 mL 3MHCl, and the reaction proceeded for another 1.5 h in the same conditions. After cooling down to room temperature, we filtered the reaction mixture through celite. We collected the desired product (TFP) through solvent extraction using 450 mL of dichloromethane. After that, we dried the mixture over anhydrous Na_2_SO_4_ and filtered. Finally, we dried the filtrate using rotary evaporator to afford 1.48 g (7.0 mmol, 14% yield) pure triformylphloroglucinol (TFP). We characterized the product by ^1^H and ^13^C NMR analyses. ^1^H NMR (400 MHz in CDCl_3_) δ 14.12 (s, 3H of -OH), δ 10.15 (s, 3H of -CHO); ^13^C NMR (100 MHz in CDCl_3_) δ 192.20 (-CHO), δ 173.73 (C attached with -OH), δ 103.05(C attached with -CHO).

### 3.5. Synthesis of TrzTFPPOP

In a simple Schiff base polycondensation reaction [47,48], we took tetraamine ligand SL-1 and tri-armed aldehyde TFP in a molar ratio of 3:4 into a 30 mL Schlenk tube with 10 mL DMSO solvent along with a magnetic stir bar. Then, we allowed the reaction mixture to stir for 10–15 min at 120 °C to dissolve both the monomers. This is followed by constantly heating the reaction mixture in the Schlenk tube for 3 days. We collected deep-orange-colored precipitate by filtration through Whatmann 40 filter paper. After that, we successively washed the materials with hot DMSO, plenty of water, THF, and MeOH, and then dried under vacuum as well as hot air oven at 100 °C overnight to achieve pure TrzTFPPOP (yield 70%).

### 3.6. Cell Cytotoxicity

We performed MTT [3-(4,5-Dimethylthiazol-2-yl)-2,5-diphenyltetrazolium bromide] assay to evaluate the cell viability [49,50]. In each well of 96-well culture plate, we plated and placed 4 × 10^3^ cells in an incubator followed by treatment with different concentrations of TrzTFPPOP ranging from 0 to 10 µg/mL for 12 h. After this treatment, we added 20 µL of 5 mg/mL MTT stock solution in each well. After 4 h of incubation at 37 °C, we solubilized the resulting intracellular formazan crystals with acidic isopropanol.

### 3.7. Quantification of Apoptosis Using Annexin V-FITC Kit

We examined apoptosis through using Annexin V-FITC apoptosis detection kit (Calbiochem, CA, USA) [51,52]. After treatment of TrzTFPPOP (3 μg/mL as per IC50) for 12 h and 24 h, we washed and stained cells with Annexin V-FITC and propidium iodide (PI) in accordance with the manufacturer’s instructions. We took 10^6^ cells for each sample analysis.

### 3.8. Quantification of Nuclear DNA in Different Phases of the Cell Cycle

We assayed cell-cycle arrest by DNA quantification [53,54]. We prepared HCT 116 cells with 3 μg/mL TrzTFPPOP for 12 h and 24 h durations. After the treatment, we fixed the harvested cells overnight in 70% cold ethanol at 4 °C followed by centrifugation, resuspension in phosphate buffer saline (PBS) containing 25 μg/mL RNase, and incubation for 1 h at 37 °C. Then, we stained cells with 50 μg/mL propidium iodide (PI) for 15 min at 4 °C. We took 10^6^ cells for each sample analysis.

### 3.9. Confocal Microscopy

The recent report says that p53, p21, and Gamma H2AX play a great role in DNA damage. So, we estimated the expression of the above proteins by confocal microscopy [55]. Briefly, after treatment with TrzTFPPOP for 12 h and 24 h, we twice washed the cover slips containing HCT 116 cells for 10 min each in 0.01 M PBS and incubated for 1 h in blocking solution containing 2% normal bovine serum and 0.3% Triton X-100 in PBS. After blocking, we incubated the slides overnight at 4 °C with the proper primary antibody (Gamma H2AX, Phospho-p53Ser46, and p21Waf1/Cip1). We used Alexa Fluor 647-tagged Gamma H2AX as the primary tagged antibody and, in case of Phospho-p53Ser46 and p21Waf1/Cip1, we used FITC- and PE-tagged anti-rabbit secondary, respectively. We diluted secondary antibodies 1:100 in blocking solution and incubated for 2 h. We then counterstained the slides with 6-diamidino-2-phenylindole (DAPI) for 10 min and mounted with the prolong anti-fade reagent (Molecular Probe, Eugene, OR, USA).

### 3.10. Statistical Analysis

We presented data as mean ±SEM (standard error of mean). We assessed statistical significance and differences among the groups via one-way analysis of variance (ANOVA) using OriginPro 8.0 software (San Diego, CA, USA). We analyzed flow cytometry by FlowJo software, which analyzed statistical data. We considered data statistically significant when *p* values were <0.05. We conducted all the experiments in triplicate to understand the consistency in the processes.

## 4. Conclusions

Here, we have demonstrated the synthesis of a secondary amine and imine-linked polyphenolic porous organic polymer TrzTFPPOP, which possesses a very high BET surface area of 2140 m^2^∙g^−1^ along with hierarchical nanoscale porosity. This highly porous organic polymer material was successfully utilized as an efficient anticancer agent towards in vitro colon cancer cell destruction through the DNA damage pathway via p53 axis. Our experimental data suggested that this porous organic polymer could involve the apoptosis mechanism through G2/M arrest via the phosphorylation of γ-H2AX. Thus, TrzTFPPOP has a very promising future as a potent anticancer agent in the treatment of colon cancer.

## Data Availability

The raw/processed data required to reproduce these findings cannot be shared at this time due to legal or ethical reasons. Samples are available from the authors.

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
