# Peer review of "N-Rich, Polyphenolic Porous Organic Polymer and Its In Vitro Anticancer Activity on Colorectal Cancer"

_molecules, 2022, doi:10.3390/molecules27217326_

Round 1

Reviewer 1 Report

The authors prepared triazine-based polyphenolic porous materials as an anti-cancer drug. The porous material was determined using IR, NMR, and TEM measurements. The anticancer activity was carefully observed. Overall, the experiments described in this paper were performed carefully. However, several points should be rewritten and discussed more carefully before publishing in Molecules.

-       Please explain about 5FU, otherwise readers do not understand.

-       What is the weight-averaged and number-averaged particle size?

-       If the sample solution is opaque due to the large particle size of the porous material, the UV data is meaningless because most of UV must not be transmitted.

-       What is the internalization pathway of the porous material into the cancer cell?

Author Response

Reviewer: 1

Please explain about 5FU, otherwise readers do not understand.

Reply:  We have defined 5-Fluorouracil (5FU) in this revised manuscript following the suggestion of the reviewer (highlighted, line 175).

-       What is the weight-averaged and number-averaged particle size?

Reply: Please note that here we have synthesized a porous polymer network through random polymerisation reaction. So, we are unable to calculate the molecular weight of the material.  That is why we have mentioned the quantity of the material was taken in µg/mL. From the HRTEM study we found that the morphology of the material was self-assemble nano wire like morphology of very small size particles. The diameter of nanowires was about 8-15 nm, which is suitable for good cellular uptake [1].

[1]  F. Zhao, Y. Zhao, Y. Liu, X. Chang, C. Chen, and Y. Zhao, small, 2011, 7, 1322-1337.

-       If the sample solution is opaque due to the large particle size of the porous material, the UV data is meaningless because most of UV must not be transmitted.

Reply: Please note that we have provided the solid sate UV-Vis spectrum to check the absorbance region of the material. Thus, no solvent was used during the data collection of UV-Vis spectrum.

-       What is the internalization pathway of the porous material into the cancer cell?

Reply: We thank the reviewer for this comment. We have added the internalization pathway of the porous material into the cancer cell in this revised version of the manuscript following this suggestion (pages 6-7).

Reviewer 2 Report

The work has some advantages, but from my point of view it requires corrections before publishing.

Here are some suggestions:

Title: The second part of the title is incorrect.

Abstract: The Abstract needs to be corrected - the first four sentences are an introduction rather than an abstract.

Cytotoxicity studies:

A.      It is also necessary to carry out cytotoxicity tests on normal not cancerous cells

B.      The cytotoxicity of TrzTFPPOP should be compared with that of a reference anticancer drug

Lines 310-311: there is: 4x103 cells, should be: 4x103 cells.

Text formatting should be carefully checked.

The language should be modified carefully.

Author Response

Reviewer: 2

The work has some advantages, but from my point of view it requires corrections before publishing.

Here are some suggestions:

Title: The second part of the title is incorrect.

Reply: We are thankful to the reviewer for this comment. We have revised the title following this suggestion of the reviewer.

Abstract: The Abstract needs to be corrected - the first four sentences are an introduction rather than an abstract.

Reply: We are thankful to the reviewer for this comment. We have revised four lines of the manuscript following this comment.

Cytotoxicity studies:

  1. It is also necessary to carry out cytotoxicity tests on normal not cancerous cells

Reply: We have carried out the cytotoxicity tests on normal cells and added the data in our revised manuscript.

  1. The cytotoxicity of TrzTFPPOP should be compared with that of a reference anticancer drug

Reply: We have added cytotoxicity data of TrzTFPPOP for the reference anticancer drug 5-Fluorouracil (5FU) in this revised manuscript following your suggestion.

Lines 310-311: there is: 4x103 cells, should be: 4x103 cells.

Reply: We have made this correction in this revised manuscript.

Text formatting should be carefully checked.

Reply: Yes, we have checked the formatting of the texts in this revised manuscript following the suggestion of the reviewer.

The language should be modified carefully.

Reply: We have carefully checked the English language of the entire manuscript and made necessary correction.

Round 2

Reviewer 2 Report

Manuscript has been improved